# Association of Peripheral Inflammatory Biomarkers and Growth Factors Levels with Sex, Therapy and Other Clinical Factors in Schizophrenia and Patient Stratification Based on These Data

**DOI:** 10.3390/brainsci13050836

**Published:** 2023-05-22

**Authors:** Evgeny A. Ermakov, Mark M. Melamud, Anastasiia S. Boiko, Daria A. Kamaeva, Svetlana A. Ivanova, Georgy A. Nevinsky, Valentina N. Buneva

**Affiliations:** 1Institute of Chemical Biology and Fundamental Medicine, Siberian Branch of the Russian Academy of Sciences, 630090 Novosibirsk, Russianevinsky@niboch.nsc.ru (G.A.N.); buneva@niboch.nsc.ru (V.N.B.); 2Department of Natural Sciences, Novosibirsk State University, 630090 Novosibirsk, Russia; 3Mental Health Research Institute, Tomsk National Research Medical Center of the Russian Academy of Sciences, 634014 Tomsk, Russiaivanovaniipz@gmail.com (S.A.I.)

**Keywords:** schizophrenia, inflammation, cytokines, growth factors, neurotrophic factors, sex differences, treatment, antipsychotics, cluster analysis, stratification

## Abstract

Multiple lines of evidence are known to confirm the pro-inflammatory state of some patients with schizophrenia and the involvement of inflammatory mechanisms in the pathogenesis of psychosis. The concentration of peripheral biomarkers is associated with the severity of inflammation and can be used for patient stratification. Here, we analyzed changes in serum concentrations of cytokines (IL-1β, IL-2, IL-4, IL-6, IL-10, IL-21, APRIL, BAFF, PBEF/Visfatin, IFN-α, and TNF-α) and growth/neurotrophic factors (GM-CSF, NRG1-β1, NGF-β, and GDNF) in patients with schizophrenia in an exacerbation phase. IL-1β, IL-2, IL-4, IL-6, BAFF, IFN-α, GM-CSF, NRG1-β1, and GDNF increased but TNF-α and NGF-β decreased in schizophrenia compared to healthy individuals. Subgroup analysis revealed the effect of sex, prevalent symptoms, and type of antipsychotic therapy on biomarker levels. Females, patients with predominantly negative symptoms, and those taking atypical antipsychotics had a more pro-inflammatory phenotype. Using cluster analysis, we classified participants into “high” and “low inflammation” subgroups. However, no differences were found in the clinical data of patients in these subgroups. Nevertheless, more patients (17% to 25.5%) than healthy donors (8.6% to 14.3%) had evidence of a pro-inflammatory condition depending on the clustering approach used. Such patients may benefit from personalized anti-inflammatory therapy.

## 1. Introduction

Schizophrenia is one of the most severe mental disorders, and its etiology and pathogenesis are not yet fully understood. According to the International Classification of Diseases, Tenth Revision (ICD-10), schizophrenia is a mental disorder characterized by at least two of the following symptoms: delusions, hallucinations, disorganized speech, grossly disorganized or catatonic behavior, or negative symptoms such as flattened affect or lack of motivation [1]. Positive, cognitive, and negative symptoms in schizophrenia may have different neurobiological bases [2] and the heterogeneity of clinical phenotypes prevents a complete understanding of its pathophysiology [3]. Similar manifestations are also observed in schizophrenia spectrum disorders, such as acute polymorphic psychotic disorder and schizotypal disorder, which are characterized by an acute onset of psychotic symptoms, but unlike schizophrenia, these manifestations are rapidly reduced and are often associated with stressful events.

It is widely accepted that schizophrenia is associated with immune dysregulation and inflammation [4,5]. Numerous studies describe pro-inflammatory changes at the molecular, cellular, and organ levels in patients with schizophrenia [4,6,7,8,9]. For example, blood and cerebrospinal fluid concentrations of pro-inflammatory cytokines, including interleukin (IL)-6, IL-8, interferon gamma (IFNγ), tumor necrosis factor alpha (TNF-α), and others, are increased in schizophrenia [6,10,11]. However, there are a number of gaps in knowledge regarding the role of inflammation in the pathogenesis of schizophrenia. First, not all cytokines have been sufficiently studied. For example, for APRIL (a proliferation-inducing ligand) and BAFF (B-cell activating factor), there is currently only one work that describes their concentration in the blood of patients with schizophrenia [12]. However, the investigation of these cytokines is interesting, since they directly affect the survival of immune cells. Secondly, sex differences in a group of patients are often not taken into account. In recent work, it has been shown that females with schizophrenia have a more pro-inflammatory profile than males [13]. Therefore, new studies of changes in cytokine levels with the analysis of clinical correlates in schizophrenia will help to fill these knowledge gaps.

Changes in concentrations of growth and neurotrophic factors are also associated with neuroinflammation and the pathogenesis of schizophrenia [14]. BDNF (brain-derived neurotrophic factor) is one of the most-studied neurotrophic factors in schizophrenia. However, data on changes in the levels of other growth/neurotrophic factors in schizophrenia are scarce. Growth factors such as GM-CSF (granulocyte macrophage colony-stimulating factor), NRG1-β1 (neuregulin-1 beta 1), NGF-β (nerve growth factor beta), and GDNF (glial cell line-derived neurotrophic factor) are actively involved in inflammation processes and schizophrenia pathogenesis [15,16,17,18]. There is a limited number of works devoted to the analysis of these growth/neurotrophic factors in schizophrenia [8,19,20]. Thus, further research on growth factors in schizophrenia is needed.

The severity of inflammation is uneven among patients with schizophrenia. Recent studies indicate that a pro-inflammatory state occurs in only 30–40% of patients with schizophrenia [21,22,23,24,25]. Moreover, the pro-inflammatory state in such subgroups of patients is associated with clinical, cognitive, and neuroanatomical anomalies. For example, patients with signs of high inflammation had worse cognitive measures and elevated thalamus, putamen, amygdala, and hippocampal volumes compared to patients with low inflammatory marker levels [21]. Thus, patients with schizophrenia can be classified into two subgroups (“high” and “low” inflammation) characterized by distinct clinical and neuroanatomical phenotypes. Therefore, the development of approaches to stratify patients based on inflammatory biomarkers is urgently needed.

This study investigated serum inflammatory and neurotrophic markers in schizophrenia to gain further insights into the role of immune and growth factors in schizophrenia pathogenesis and heterogeneity. Therefore, the current study aimed to (1) analyze the changes in a panel of cytokines and growth factors in patients with schizophrenia compared to healthy controls, (2) explore the effects of clinical variables on biomarker levels, and (3) stratify patients based on inflammatory markers and compare their clinical profiles.

## 2. Materials and Methods

### 2.1. Study Design

This work is a cross-sectional and case-control study. In the first part of the study, the concentrations of cytokines and growth factors were compared in groups of patients with schizophrenia and healthy individuals. In the second part of the study, a subgroup analysis of indicators in the analyzed groups was carried out. Cluster analysis was then performed to identify two subgroups (with and without severe inflammation). This study was conducted in accordance with the Declaration of Helsinki, and approved by the Local Ethics Committee of the Institute of Chemical Biology and Fundamental Medicine (protocol N8 from 7 February 2020). Informed consent was obtained from all subjects involved in the study.

### 2.2. Participants

The cohort of this study included 54 patients with schizophrenia (F20 according to the ICD-10) in an exacerbation phase and 40 healthy individuals. The healthy donors and patients were recruited at the Mental Health Research Institute of the Tomsk National Research Medical Center (Tomsk, Russia) between February 2020 and December 2022.

The inclusion criteria for patients were as follows:(1)Established diagnosis schizophrenia (F20.0 or F20.6 according to ICD-10);(2)Consent to participate in the study.

The exclusion criteria for all subjects were as follows:(3)History of autoimmune, oncological diseases, and infections of the central nervous system;(4)Signs of acute infectious or allergic diseases at the time of the study;(5)Neurological and somatic diseases that make it difficult to objectively assess the clinical condition;(6)Comorbid psychiatric diagnoses, substance abuse or dependence (excluding nicotine and caffeine), mental retardation;(7)Pregnancy;(8)Obesity, diabetes, or hypertension;(9)Recent surgeries and head injuries;(10)Recent vaccination against any infections.

Exclusion criteria for healthy donors were as follows:(1)Presence of relatives with psychiatric diagnosis;(2)Lack of consent to study.

It is important to note that all patients were taking antipsychotic drugs prescribed by the treating physician, including typical (haloperidol (10 people), trifluoperazin (4 people), chlorprothixene (2 people) and atypical antipsychotics (risperidone (14 people), quetiap-ine (7 people), olanzapine (6 people), clozapine (6 people)). Antipsychotic doses were converted to mean daily chlorpromazine equivalent dose (CPZeq) [26].

Psychiatric symptoms were assessed with the Positive and Negative Syndrome Scale (PANSS) [27]. The clinical status of patients was assessed by experienced clinicians. Based on PANSS scores, patients were divided into subgroups with predominantly negative and positive symptoms. Additionally, patients were also divided into subgroups depending on the response to therapy based on the consensus guidelines of the Treatment Response and Resistance to Psychosis Working Group published in 2017 [28]. Patients were classified into the treatment-resistant schizophrenia (TRS) subgroup under the following conditions: a poor response (PANSS scores > 60) to two different adequate doses (CPZeq > 600 mg per day) and course durations (>6 weeks) of non-clozapine antipsychotics. The rest of the patients were classified into the partially responsive schizophrenia (PRS) subgroup.

### 2.3. Biological Material

Blood samples were collected in a PET tube BD Vacutainer with a clot activator (silica) (cat. # 369032, BD, Franklin Lakes, NJ, USA) from each subject after eight hours of overnight fasting on the first days of hospitalization. Serum was extracted by centrifugation for 30 min at 2000× *g* and 4 °C. Serum samples were stored at –80 °C until analysis.

### 2.4. Multiplex Immunoassay of Cytokines and Growth Factors in Serum

All measurements were carried out at the Laboratory of Molecular Genetics and Biochemistry in Mental Health Research Institute, Tomsk National Research Medical Center (Tomsk NRMC). The concentrations of eleven cytokines (IL-1β, IL-2, IL-4, IL-6, IL-10, IL-21, APRIL, BAFF, PBEF/Visfatin, IFN-α, and TNF-α) and four growth/neurotrophic factors (GM-CSF, NRG1-β1, NGF-β, and GDNF) were determined on multiplex analyzers Magpix and Luminex 200 (Luminex, Austin, TX, USA) based at the Core Facility “Medical Genomics” (Tomsk NRMC). Analytes were measured using the Human Premixed Multi-Analyte Kit Luminex Assay (cat. # LXSAHM, R&D Systems, Minneapolis, MN, USA). The received signals were processed in xPONENT software (Luminex, Austin, TX, USA). The output data were exported to the MILLIPLEX Analyst 5.1 software (Merck, Darmstadt, Germany) for final analysis as in [29,30]. The final results were presented as pg/mL.

### 2.5. Data Preprocessing and Statistical Analysis

Outliers were removed if the value did not fall within the range of the calibration curve or was outside two standard deviations from the group mean. The analyzed concentrations of cytokines and growth factors had 1.1% to 4.3% of outliers, with the exception of PBEF/Visfatin, for which 21.3% of outliers were excluded.

Statistical data analysis was carried out in STATISTICA 10 (StatSoft, Tulsa, OK, USA) and OriginPro 2021 (OriginLab, Northampton, MA, USA). Each marker was tested for normality using the Shapiro–Wilk test. Most of the markers had a non-normal distribution. The significance of the differences in biomarker levels was calculated using the Mann–Whitney U test or Kruskal–Wallis test followed by Dunn’s post hoc test for multiple comparisons (when comparing more than two groups). The Benjamini–Hochberg procedure was used for multiple comparisons correction. Differences were considered significant only for variables that remained significant after Benjamini–Hochberg correction. Pearson’s chi-squared test was used to analyze the significance of the differences in categorical variables (sex, prevailing symptoms, antipsychotic type). The Spearman rank correlation coefficient was calculated to evaluate the correlation dependences.

At the preliminary stage, hierarchical clustering was used to estimate the optimal number of clusters. Euclidean distance is used as distance metric. The optimal number of clusters was two. Then, *K*-means clustering was used to classify patients and healthy individuals into two clusters/subgroups with signs of high inflammation (“High inflammation” cluster) or low inflammation (“Low inflammation” cluster). The classification was based on data from all biomarkers (with the exception of PBEF/Visfatin, which has many outliers), ten pro- and anti-inflammatory cytokines, or four growth factors. Biomarker data were median-normalized. Median normalization was performed by subtracting the median of the variable’s distribution from each value in the data sample and normalizing by the median deviation. The STATISTICA 10 software (StatSoft, Tulsa, OK, USA) was used for cluster analysis. OriginPro 2021 (OriginLab, Northampton, MA, USA) was used for plotting.

## 3. Results

### 3.1. Demographic and Clinical Characteristics of Participants

The demographic and clinical characteristics of the participants are presented in Table 1. The study included 54 patients with schizophrenia (SZ group) and 40 healthy subjects (HS group). All patients included in the study were in the acute phase of the disease. Most of included patients (81.5%) were diagnosed with paranoid schizophrenia (F20.0), and the rest of the patients (18.5%) were diagnosed with simple schizophrenia (F20.6). Age was distributed similarly between SZ and HS groups. However, the sex ratio was different. There were more females than males in the SZ group (Table 1). Therefore, we took into account these differences in further analysis. Age, disease duration, and age at disease manifestation did not differ between males and females in the SZ group. PANSS negative, PANSS general, and PANSS total scores were higher in male patients with schizophrenia than in female patients. The duration of antipsychotic therapy did not differ in the subgroup of male and female patients. Antipsychotic doses converted to mean daily CPZeq did not differ between males and females, although there was a trend towards higher mean daily CPZeq in men. In the total SZ group, there were more patients treated with atypical than typical antipsychotics. However, males were more likely to receive typical antipsychotics (Table 1).

### 3.2. Group Differences in Biomarker Levels

Serum concentrations of pro- and anti-inflammatory cytokines and growth/neurotrophic factors were determined in SZ and HS groups (Figure 1). Among cytokines, the concentration of IL-1β, IL-2, IL-4, IL-6, BAFF, and IFN-α was shown to be higher in the SZ group compared to the HS group. The concentration of TNF-α decreased in the SZ group. Levels of IL-10, IL-21, APRIL, and PBEF/Visfatin did not change statistically significantly. Importantly, the IL-10/IL-6 ratio was lower in the SZ group, which indicates the predominance of pro-inflammatory processes in schizophrenia (Figure 1L).

Among growth factors, the concentration of GM-CSF, NRG1-β1, and GDNF significantly increased in the SZ group compared to the HS group. On the contrary, NGF-β significantly decreased in the SZ group (Figure 1).

Numerical data including median, mean, and standard deviation for all analyzed biomarkers that can be used for subsequent meta-analyses are presented in Appendix A.

### 3.3. Association of Biomarker Levels with Sex, Therapy, and Prevalent Symptoms

The sex of participants can significantly influence the results of biomarker studies. Therefore, the levels of cytokines and growth factors were analyzed in subgroups of males and females (Table 2). The greatest changes in biomarker levels have been shown to occur in females. In males, only the levels of IL-6 and BAFF increased statistically significantly compared to healthy controls. However, in females, the levels of IL-1β, IL-2, IL-4, IL-6, IL-10, BAFF, and IFN-α increased significantly (Table 2). Interestingly, the concentrations of growth factors changed only in females; in males, they did not change significantly. In females, serum levels of GM-CSF, NRG1-β1, and GDNF were higher and NGF-β decreased. Thus, the observed increase in IL-1β, IL-2, IL-4, IFN-α, GM-CSF, NRG1-β1, and GDNF and the decrease in NGF-β in the total SZ group (please see Figure 1) were driven by the contributions of females. However, the levels of IL-6 and BAFF increased in a similar way in males and females, so it can be assumed that these changes are sex-independent.

A comparison of the level of biomarkers between males and females in the HS group showed that only the level of TNF-α was different (Table 2). In the SZ group, the concentrations of TNF-α and NGF-β differed in males and females. Thus, the concentration of TNF-α was higher in males than females in both the SZ group and the HS group. The serum level of NGF-β significantly increased in males only in the SZ group. Although the level of NGF-β decreased in the general SZ group compared to the HS group (Figure 1), the concentration of NGF-β appears to be higher in males than in females (Table 2).

Further, all patients were divided into subgroups with prevailing negative or positive symptoms. Among all biomarkers, only the level of IL-6 was significantly higher in patients with prevailing negative symptoms (Figure 2A). The significantly low IL-10/IL-6 ratio in patients with negative symptoms also confirmed the pro-inflammatory state in this subgroup of schizophrenic patients (Figure 2B).

Correlation analysis did not reveal clinically significant correlations with the analyzed biomarkers, with the exception of age. Serum GDNF concentration decreased with increasing age (Rs = −0.22, *p* = 0.04, Spearman correlation). This dependence is graphically represented in Appendix A. The age of the patients did not affect the concentration of other biomarkers. The duration of the disease also did not affect the level of cytokines. There were no significant differences in the level of biomarkers in the subgroups of patients with a disease duration of up to 10 and more than 10 years (Appendix A).

Further, the effect of antipsychotic therapy on the level of biomarkers in schizophrenia was studied (Table 3). In the group of patients taking second-generation antipsychotics, also known as “atypical antipsychotics”, the concentrations of IL-1β, IL-2, IL-6, IL-21, IFN-α, GM-CSF, NRG-1β1, and GDNF were significantly higher than in the group of patients on first-generation antipsychotics (“typical antipsychotics”). However, IL-10 decreased in patients taking typical antipsychotics. Interestingly, the duration of therapy (up to a year and more than a year of therapy subgroups) did not affect the level of biomarkers (Appendix A).

Additionally, patients were divided into TRS and PRS subgroups depending on the response to therapy (for details, see Section 2.2). However, no statistically significant differences in biomarker levels were found between these groups (Appendix A).

### 3.4. Cluster Analysis and Patient Stratification Based on Biomarker Data

Clustering helps to identify more homogeneous subgroups in the sample. Preliminary cluster analysis using hierarchical clustering showed that the participants are divided into two main clusters. Therefore, all participants were further divided into two clusters using K-means clustering. K-means clustering was performed based on the concentration data of 14 biomarkers (all biomarkers except for PBEF, which had many outliers). As a result, subgroups with signs of high inflammation (“High inflammation” cluster) or low inflammation (“Low inflammation” cluster) were identified in SZ and HS groups (Figure 3).

The concentration of all cytokines except APRIL was significantly higher in the “high inflammation” cluster than in the “low inflammation” cluster in the total sample (Appendix A). Descriptive statistics between two clusters in SZ and HS groups are presented in Table 4. Results of Kruskal–Wallis ANOVA test followed by Dunn’s post hoc test for multiple comparisons are shown in Appendix A. It was shown that the concentrations of all cytokines except IL-10, APRIL, and PBEF differed significantly between patients and healthy donors classified in “high” and “low inflammation” clusters (Table 4). However, differences in the clinical parameters of patients from the “low” and “high inflammation” cluster could not be identified (Table 4). In the subgroup of patients with schizophrenia with “high inflammation”, disease duration, PANSS scores, and total CPZeq did not differ significantly from those in the “low inflammation” cluster. This is partly explained by the small number of patients classified in the “high inflammation” cluster.

The ratio of males and females in the subgroups was slightly different. Therefore, the significance of differences in median levels of cytokines and growth factors in males and females in “low” and “high inflammation” cluster was studied (Appendix A). The concentrations of all biomarkers changed in the same way in males and females in the two subgroups. The only exception was IL-10, which significantly increased in females from the “high inflammation” cluster, but did not change in males.

Clustering persistence was then tested using ten (pro- and anti-inflammatory cytokines: IL-1β, IL-2, IL-4, IL-6, IL-10, IL-21, APRIL, BAFF, IFN-α, and TNF-α) and four (growth factors: GM-CSF, NRG-1β1, NGF-β, and GDNF) biomarkers. The results were similar to those obtained using 14 biomarkers (Figure 4). This pattern is also confirmed by the Venn diagrams on the intersection of samples when divided into different clusters using three clustering approaches (Appendix A).

Thus, 17% to 25.5% of patients and 8.6% to 14.3% of healthy donors were found to have significantly high levels of cytokines and growth factors.

## 4. Discussion

### 4.1. Group Differences in Biomarker Levels

The results of this work confirm the pro-inflammatory state of patients with schizophrenia, which was previously described [4,31,32]. A decrease in the IL-10/IL-6 ratio also confirms the activation of pro-inflammatory processes in schizophrenia (Figure 1L). A decreased IL-10/IL-6 ratio is known to be a robust marker of inflammation [33]. Our finding of elevated IL-6 in schizophrenia is consistent with multiple previous studies [8,10,11,34,35]. In contrast, we found decreased TNF-α levels, unlike most other studies reporting no change or increased TNF-α [10,36,37]. The decrease in TNF-α level in our sample may be explained by the higher proportion of males, where TNF-α was significantly lower (Table 2). The level of anti-inflammatory cytokine IL-10 did not change (Figure 1), which is also consistent with a number of studies [36,38]. However, it should be noted that a number of studies have found an increase in the concentration of IL-10 in patients with schizophrenia [8,13]. GM-CSF released from many cells in response to inflammatory mediators (e.g., IL-6) was also higher in patients with schizophrenia, consistent with previous studies [8]. An increase in the level of IL-4 (Figure 1C), also described in a number of other works [8,36,37], may be of a compensatory nature, since this cytokine has anti-inflammatory functions [13]. We also found an increase in the serum levels of IL-1β in patients with schizophrenia (Figure 1A), which is also confirmed by a number of studies [34,37,39,40,41]. However, there are a number of studies that describe the absence of changes in the concentrations of IL-1β in schizophrenia [8,13,36,42]. We also found an increase in serum IL-2 concentration in schizophrenia (Figure 1B). It is known that IL-2 may play a role in the pathogenesis of schizophrenia in the context of the T-reg hypothesis [43]. However, studies of the concentration of IL-2 in the blood of patients show conflicting results [43].

This study also revealed changes in the concentrations of some cytokines that are rarely studied in schizophrenia. In particular, an increase in the level of BAFF (Figure 1H) and IFN-α (Figure 1J) in the serum of patients with schizophrenia compared to healthy individuals was shown. An increased level of BAFF in schizophrenia indicates dysregulation of humoral immunity in schizophrenia, since this cytokine regulates the growth and proliferation of B cells. Serum levels of PBEF/Visfatin, IL-21, and APRIL showed no significant difference between patients and healthy subjects (Figure 1). A few literature data indicate that the levels of BAFF and IFN-α do not change in schizophrenia [12,13], which is inconsistent with our data. There are studies that have not found differences in PBEF/Visfatin and IL-21 levels [44,45], similar to our results. One study showed a decrease in the concentration of APRIL [12]. Our results add to the knowledge about the levels of these biomarkers in schizophrenia.

Among the growth/neurotrophic factors, we observed a decrease in the concentration of NGF-β and an increase in GM-CSF, NRG-1β1, and GDNF (Figure 1). An increase in GM-CSF has been identified previously in other studies [8,46]. Data on the decrease in NGF-β are consistent with previous studies showing a decrease or no change in the level of this neurotrophin in schizophrenia [19,47,48,49,50]. The level of NRG-1β1 has been studied less in schizophrenia. Two studies have shown a decrease in the concentration of NRG-1β1 in the serum of patients [20,51]. However, our data on the increase in NRG-1β1 are explained by the effect of antipsychotic therapy, since the therapy has been shown to increase the level of NRG-1β1 [51]. Our results on the increase in GDNF in schizophrenia do not agree with the literature data. Previous studies have shown no change or decreased levels of this neurotrophic factor in schizophrenia [19,48,52,53,54,55]. However, our data may be explained by the effect of therapy, as there is evidence that GDNF may be increased after treatment [55].

### 4.2. Demographic and Clinical Correlates

The prevailing positive or negative symptomatology of schizophrenia may be associated with changes in cytokine levels. We showed that in patients with predominant negative symptoms, the level of IL-6 was significantly higher than in patients with positive symptoms (Figure 2A). The correlation between a negative PANSS score and IL-6 has already been described earlier in a number of papers [56,57]. We also found that the IL-10/IL-6 ratio was significantly lower in patients with negative symptoms (Figure 2B). As stated above, a decrease in this ratio is a clear sign of inflammation. There is evidence that an increase in inflammatory markers, particularly IL-6, is a poor prognostic sign, since patients with severe negative symptoms and a pro-inflammatory state are much less treatable [58].

This work also demonstrates the effect of gender on changes in the levels of cytokines and growth factors (Table 2). Females had a much more pro-inflammatory profile than males. In females, the concentrations of IL-1β, IL-2, IL-4, IL-6, IL-10, BAFF, IFN-α, and GM-CSF increased, while in males, only IL-6 and BAFF increased. The revealed decrease in the level of TNF-α in the general group may also be gender-dependent. Most studies describe an increase in TNF-α in the blood in schizophrenia [10,13,36] or no changes [8,37]. Our work showed that TNF-α significantly decreased in males, while it did not change in females. Overall, TNF-α levels were significantly higher in males than in females. Therefore, the decrease in the level of TNF-α in the general group is associated with the influence of the male group, although antipsychotic therapy can also reduce the concentration of TNF-α in the blood [59]. Our findings on the gender difference in cytokine levels are consistent with the recent work by Mednova et al. [13], which also describes a more pro-inflammatory cytokine profile in females with schizophrenia than in males.

Different types of antipsychotic drugs can affect cytokine levels in different ways. We showed that in the group of patients taking atypical antipsychotics, the level of pro-inflammatory cytokines was higher than in patients on typical antipsychotics. In our study, among patients taking atypical antipsychotics, 42% received risperidone and 58% received quetiapine, olanzapine, or clozapine. It is believed that atypical antipsychotics have anti-inflammatory properties [60]. However, there are works describing the pro-inflammatory effect of some atypical antipsychotics, such as clozapine [61,62]. Melbourne et al. (2020) showed that risperidone and protracted illness were associated with an increase in JAK-STAT1 signature in peripheral blood mononuclear cells, shifting them to a pro-inflammatory phenotype [63]. We have also previously shown that metabolic syndrome can impact multidirectional changes in cytokine levels in patients receiving the same therapy with different atypical antipsychotics [64]. In particular, treatment with risperidone, as the least atypical antipsychotic, led to an increase in pro-inflammatory IFN-α2, IL-1β, and IL-7 in schizophrenia patients with metabolic syndrome and was ineffective in lowering pro-inflammatory cytokines in patients without it. In contrast to the anti-inflammatory effect of many atypical antipsychotics in first administration [46] or short-term drug therapy [65], the extended usage of antipsychotic drugs might lead to metabolic disorders and an increase in inflammatory markers in some cases [46,66,67]. Vassilopoulou E. et al., in the long-term study of side effects during 5 years of atypical antipsychotic treatment, indicated their heterogeneity in metabolic and inflammatory effects [68].

According to in vitro studies, typical antipsychotics, including haloperidol and clopromazine, also have anti-inflammatory properties [69,70,71]. Obuchowicz E. et al. have demonstrated that chlorpromazine and haloperidol, dependently on initial glia activation status via different mechanisms, may induce anti-inflammatory effects and decrease the production of pro-inflammatory cytokines [72]. Therefore, the observed pro-inflammatory effects of atypical antipsychotics may be related to the effects of some atypical antipsychotics.

Neurotrophins and antipsychotics are the main modulators of neuronal activity. It is expected that antipsychotic drugs, modulating neuronal activity, affect neurotrophic fac-tor-related pathways [73,74]. A number of studies report that atypical antipsychotics have more pronounced neuroprotective effects than typical anti-psychotics. Pillai A. et al. showed that 90-day treatment with typical antipsychotics led to a 2-fold reduction in NGF levels in the hippocampus of rats compared to atypical antipsychotics. Switching to atypical antipsychotics (risperidon, olanzapine) for the next 90 days significantly restored levels of NGF and BDNF [75]. Atypical antipsychotics olanzapine and lurasidone lead to an increase in serum NGF levels [76]. Serum NRG-1β1 levels increased significantly following atypical antipsychotic treatment, but nonresponders showed no such effect in Yang H. et al.’s study [51], and NRG1 mRNA expression increased after atypical antipsychotic treatment [77], demonstrating that NRG1 may serve as a potential therapeutic marker. Several studies support that treatment with atypical antipsychotics increases serum GDNF levels after therapy [78], especially in responder subgroups [55]. Thus, the results of our research indicating significantly higher levels of neurotrophic factors NRG1β1 and GDNF in patients taking atypical antipsychotics are consistent with existing literature; however, we were unable to identify differences in NGF levels.

We did not find significant associations between biomarker levels and age, illness duration, or PANSS scores, possibly due to the limited sample size. The lack of differences between the TRS and PRS subgroups of patients can also be explained by the small sample size of the TRS group (9 persons). However, patients with predominantly negative symptoms showed higher IL-6 and lower IL-10/IL-6 ratios, indicating a link between inflammation and negative symptom severity that warrants further investigation.

### 4.3. Patient Stratification by Inflammatory Biomarkers

Recent studies have shown that an increase in peripheral or neuroinflammation is observed in only 30–40% of patients with schizophrenia [21,22,23,24]. Thus, subgroups of “high” and “low inflammation” can be identified among patients. Using cluster analysis, we showed that 17% to 25.5% of patients and 8.6% to 14.3% of healthy donors had evidence of “high inflammation” (Figure 4). We used three sets of biomarkers to construct the classification: 14 biomarkers (cytokines and growth factors), 10 biomarkers (cytokines only), and 4 biomarkers (growth/neurotrophic factors only). As a result, similar “high” and “low inflammation” subgroups were identified in patients and healthy donors. Our data confirm the existence of a subgroup of schizophrenic patients with severe inflammation. The “high inflammation” cluster comprised around 25% of patients, with significantly higher levels of both pro-inflammatory cytokines as well as certain growth factors such as NRG-1β1 (Table 4). Therefore, in further studies, the classification of patients can be based on data on the level of cytokines and growth factors. Despite the differences described above between patients taking different classes of antipsychotics, we believe this did not affect the distribution of patients into clusters. Both the low- and high-inflammation clusters contain approximately two-thirds of patients taking atypical antipsychotics. The portions of patients taking typical antipsychotics were 21% and 34% in the low- and high-inflammation clusters, respectively, representing comparable proportions. Though clinical parameters (disease duration, PANSS scores, and total CPZeq) did not differ significantly between clusters, the “high inflammation” group may represent an inflammatory endophenotype of schizophrenia with distinct pathophysiology and treatment needs. The lack of differences in clinical parameters is possibly due to the small number of individuals classified in the “high inflammation” subgroup. Additionally, patients from the “high inflammation” subgroup may have cognitive and neuroanatomical abnormalities, which were not studied in this research. However, these results may have translational significance. Patients in the “high inflammation” subgroup can be personally recommended anti-inflammatory therapy in addition to antipsychotics. Further research should focus on developing stratification criteria for patients with schizophrenia based on inflammatory biomarkers to recommend anti-inflammatory therapy for patients in the “high inflammation” subgroup.

### 4.4. Significance of Findings, Limitations, and Directions for Future Research

Our study provides further evidence of immune system dysregulation and inflammatory abnormalities in a subgroup of schizophrenia patients. The identification of inflammatory biomarkers may aid in diagnosis, prognostication, and guiding personalized treatment approaches. The results of this study should be interpreted with caution. The main limitation is related to the small sample of participants. Additionally, the potential effects of endocrine and metabolic disturbances, analgesics, smoking, and caffeine were not considered in our data analysis. Larger longitudinal studies are needed to confirm these findings and determine how inflammatory profiles may relate to disease manifestation, severity, treatment response, and cognition. Further studies with a larger sample size of patients receiving monotherapy are also needed for a more detailed analysis of the effects of antipsychotic therapy on the levels of cytokines and neurotrophic and growth factors.

## 5. Conclusions

This study demonstrated changes in serum concentrations of cytokines and growth factors in patients with schizophrenia compared with healthy donors. Significantly higher levels of the pro-inflammatory cytokine IL-6 and lower levels of the anti-inflammatory IL-10 have been found in patients with schizophrenia, indicating a pro-inflammatory state. Among growth factors, changes in GM-CSF, NRG1-β1, NGF-β, and GDNF were revealed. It has been shown that the inflammatory profile in patients with schizophrenia was sex-dependent. Females were characterized by a more severe pro-inflammatory phenotype. It was also demonstrated that the concentration of the pro-inflammatory cytokine IL-6 was increased and the ratio of IL-10/IL-6 was decreased in patients with negative symptoms compared to patients with positive symptoms. It was found that patients treated with atypical antipsychotics showed an increase in pro-inflammatory cytokines and growth factors. Using cluster analysis, we identified a subgroup of patients with a pro-inflammatory phenotype. Patients in this subgroup may be recommended personalized anti-inflammatory therapy.

## Figures and Tables

**Figure 1 brainsci-13-00836-f001:**
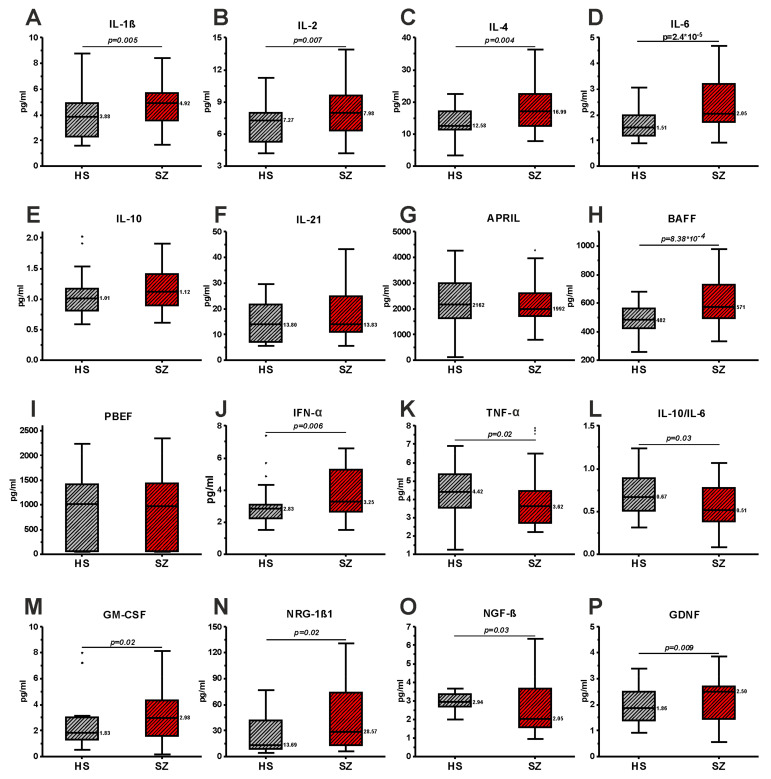
Serum levels of IL-1β (**A**), IL-2 (**B**), IL-4 (**C**), IL-6 (**D**), IL-10 (**E**), IL-21 (**F**), APRIL (**G**), BAFF (**H**), PBEF/Visfatin (**I**), IFN-α (**J**), TNF-α (**K**), GM-CSF (**M**), NRG1-β1 (**N**), NGF-β (**O**), GDNF (**P**), and IL-10/IL-6 ratio (**L**) in patients with schizophrenia and healthy individuals. Outliers identified by Tukey’s test are shown as black dots. The figure shows only statistically significant differences (*p*-values) calculated using the Mann–Whitney U test with Benjamini–Hochberg correction for multiple comparisons.

**Figure 2 brainsci-13-00836-f002:**
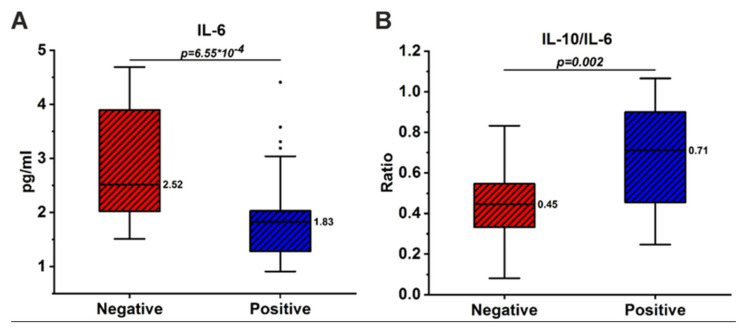
IL-6 concentration (**A**) and IL-10/IL-6 ratio (**B**) in patients with schizophrenia depending on the prevailing symptoms. Outliers identified by Tukey’s test are shown as black dots. The Mann–Whitney test was used to assess the significance of differences (*p*-values).

**Figure 3 brainsci-13-00836-f003:**
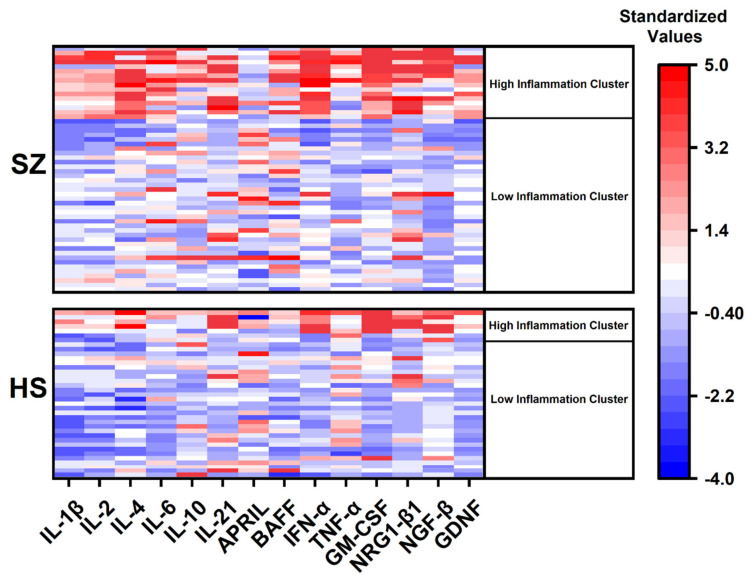
Heatmap and cluster analysis of biomarker data in SZ and HS groups. Heatmap colors reflect normalized data on biomarker levels. Each row represents a specific participant. Using K-means clustering based on concentration data of 14 biomarkers, subgroups with signs of high inflammation (“High inflammation” cluster) or low inflammation (“Low inflammation” cluster) were identified in SZ and HS groups.

**Figure 4 brainsci-13-00836-f004:**
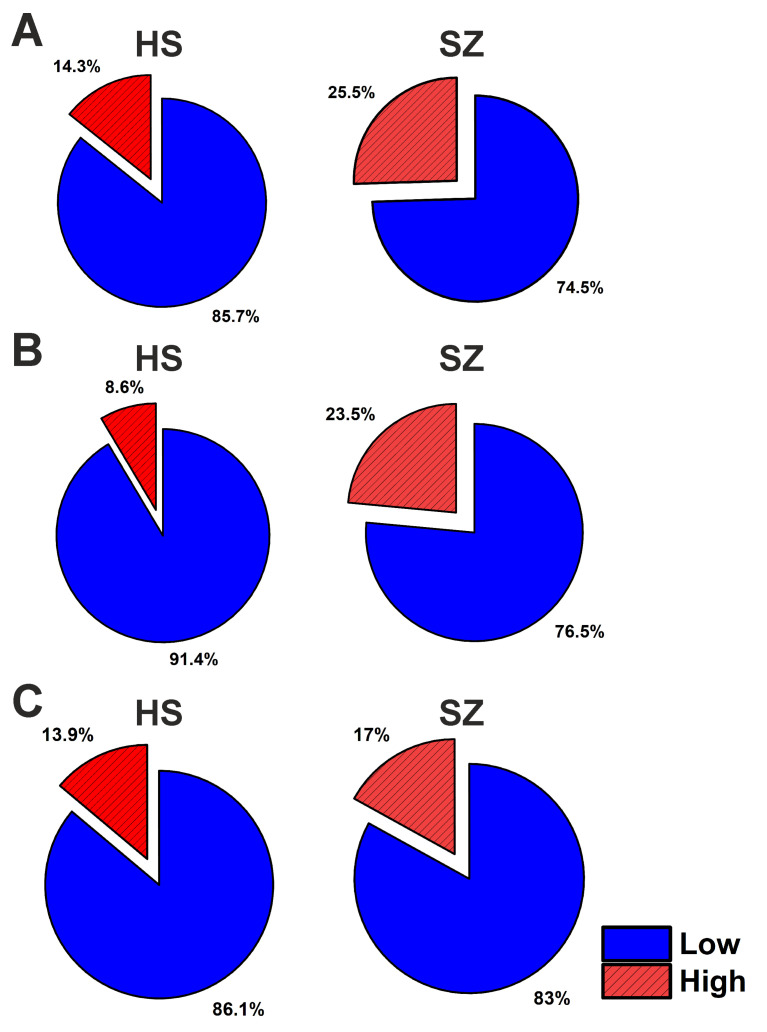
Percentage of participants classified in “Low” and “High inflammation” clusters using data from 14 (**A**), 10 (**B**), and 4 (**C**) biomarkers. The list of biomarkers used for clustering is given in the text.

**Table 1 brainsci-13-00836-t001:** Basic clinical and demographic characteristics of healthy individuals and patients with schizophrenia included in the study.

Variable	HS Group (*n* = 40)	SZ Group (*n* = 54)	SZ Female Group (*n* = 40)	SZ Male Group (*n* = 14)	*p*-Value	
					HS vs. SZ	SZ Female vs. SZ Male
Age, years	36.5 (28–58.8)	39 (33–51)	39 (33–50.3)	39 (31.8–52.3)	0.72	0.83
Sex (F/M), %	50/50	74/26	-	-	0.02 *	-
Age at disease manifestation, years	-	25 (20–31)	26 (20–31)	20.5 (18.5–30.3)	-	0.35
Disease duration, years	-	15 (9–18)	15 (7–17)	16.5 (14.8–21.5)	-	0.15
Prevailing symptoms (positive/negative), %	-	54/46	58/42	42/58	-	0.68
PANSS, positive score	-	17 (13–24)	15 (12–21)	23.5 (15–26)	-	0.1
PANSS, negative score	-	22 (17–27)	20 (16–25)	27.5 (19.8–29)	-	0.02 *
PANSS, general score	-	40 (33–54)	37 (32–49)	55.5 (37–61.3)	-	0.03 *
PANSS, total score	-	78 (65–106)	75 (64–92)	109 (71–115.3)	-	0.02 *
Total CPZeq	-	306.3 (200–580.9)	300 (200–532)	500 (225–783)	-	0.23
Antipsychotic type (atypical/typical/ polytherapy), %	-	63/31/6	70/23/8	38/54/8	-	0.03 *
Duration of antipsychotic therapy, years	-	10 (2–16)	10 (1.35–16)	10 (3.5–16)	-	0.63

Note: Data are presented as a median (Q1–Q3). The significance of the differences was calculated using the Mann–Whitney U test or Pearson’s chi-squared test (for categorical variables). * Statistically significant differences. PANSS: Positive and Negative Syndrome Scale; CPZeq: daily chlorpromazine equivalents.

**Table 2 brainsci-13-00836-t002:** Serum cytokine and growth factor levels of patients with schizophrenia and healthy individuals depending on sex.

Variable	Females			Males			Females vs. Males *p*-Value	
	HS Group (*n* = 19)	SZ Group (*n* = 40)	*p*-Value	HS Group (*n* = 19)	SZ Group (*n* = 14)	*p*-Value	HS Group	SZ Group
Cytokines								
IL-1β	3.59 (1.67–4.92)	4.93 (4.2–5.7)	0.002 *	4.22 (2.48–5.62)	3.59 (2.57–6.44)	0.59	0.41	0.13
IL-2	7.04 (5.21–7.88)	8.31 (6.54–9.61)	0.004 *	7.46 (5.29–8.63)	6.42 (5.81–10.02)	0.53	0.35	0.64
IL-4	12.52 (10.55–17.45)	16.98 (12.51–22.53)	0.01 *	15.21 (7.97–17.01)	17.93 (12.24–29.02)	0.12	0.48	0.79
IL-6	1.26 (0.94–2.05)	2.02 (1.71–3.15)	0.0004 *	1.53 (1.28–1.99)	2.33 (1.75–3.66)	0.017 *	0.11	0.5
IL-10	0.99 (0.78–1.05)	1.08 (0.9–1.37)	0.04 *	1.12 (0.92–1.39)	1.19 (0.83–1.6)	0.49	0.06	0.38
IL-21	17.03 (7.19–29.68)	13.83 (10.91–24.82)	0.88	10.27 (7.15–16.86)	13.83 (8.31–33.73)	0.3	0.08	0.69
APRIL	2003 (1498–3016)	1996 (1625–2699)	0.78	2332 (1919–3005)	1979 (1801–2421)	0.22	0.22	0.85
BAFF	481 (405–564)	557 (467–749)	0.04 *	486 (441–575)	609 (515–698)	0.002 *	0.8	0.52
IFN-α	2.81 (2.26–3.06)	3.25 (2.75–4.95)	0.01 *	2.85 (2.2–3.65)	3.56 (2.1–5.37)	0.38	0.9	0.93
TNF-α	3.67 (2.93–5.25)	3.38 (2.66–4.35)	0.29	4.74 (4.13–5.64)	4.26 (3.44–4.85)	0.11	0.03 *	0.03 *
Growth/neurotrophic factors								
GM-CSF	1.63 (1.04–2.26)	2.98 (1.66–4.16)	0.01 *	2.45 (1.29–3.17)	3.32 (1.59–7.42)	0.24	0.34	0.73
NRG-1β1	25.62 (6.38–59.35)	28.57 (13.48–75.91)	0.14	13.43 (8.53–42.24)	36.27 (12.4–78.67)	0.16	0.62	0.69
NGF-β	2.82 (2.46–3.23)	1.83 (1.53–3.34)	0.03 *	3.25 (2.71–4.65)	3.81 (1.79–6.43)	0.96	0.18	0.03 *
GDNF	1.45 (1.35–2.21)	2.55 (1.86–2.69)	0.002 *	2.14 (1.44–2.59)	1.87 (1.39–3.07)	0.97	0.16	0.29

Note: Data are presented as a median (Q1–Q3), pg/mL. The significance of the differences was calculated using the Mann–Whitney U test. * Statistically significant differences after Benjamini–Hochberg correction for multiple comparisons.

**Table 3 brainsci-13-00836-t003:** Serum cytokine and growth factor levels of patients with schizophrenia depending on the type of antipsychotic used for therapy.

Variable	Patient Group on Typical Antipsychotics (*n* = 16)	Patient Group on Atypical Antipsychotics (*n* = 33)	*p*-Value
Cytokines			
IL-1β	3.06 (2.54–4.75)	4.95 (4.89–6.49)	0.002 *
IL-2	6.34 (5.26–7.91)	8.71 (7.32–10.34)	0.006 *
IL-4	14.96 (11.83–20.7)	18.87 (15.21–31.2)	0.052
IL-6	1.65 (1.26–2.73)	2.25 (1.96–3.31)	0.009 *
IL-10	1.43 (1.1–1.68)	1.04 (0.79–1.37)	0.01 *
IL-21	10.59 (7.18–13.73)	16.86 (12.31)	0.003 *
APRIL	1847 (1572–2579)	2124 (1806–2756)	0.23
BAFF	537 (467–636)	577 (494–809)	0.15
IFN-α	2.66 (1.9–3.6)	3.47 (3.12–5.66)	0.01 *
TNF-α	3.46 (2.77–4.77)	3.82 (2.66–4.55)	0.94
Growth/neurotrophic factors			
GM-CSF	2.06 (1.16–3.07)	3.03 (2.05–4.7)	0.04 *
NRG-1β1	18.72 (11.54–45.37)	42.24 (17.06–120.56)	0.04 *
NGF-β	2.55 (1.6–4.59)	2.04 (1.55–3.83)	0.95
GDNF	1.39 (1.04–2.11)	2.6 (2.23–3.28)	0.0002 *

Note: Data are presented as a median (Q1–Q3), pg/mL. The significance of the differences was calculated using the Mann–Whitney U test. * statistically significant differences after Benjamini–Hochberg correction for multiple comparisons.

**Table 4 brainsci-13-00836-t004:** Descriptive statistics between two clusters in healthy individuals and patients with schizophrenia.

Cluster	“Low Inflammation” Cluster		“High Inflammation” Cluster		*p*-Value
Group	HS (*n* = 30)	SZ (*n* = 38)	HS (*n* = 5)	SZ (*n* = 14)	
IL-1β	2.57	4.90	6.42	7.23	1.71 × 10^−7^ *
IL-2	7.03	7.74	8.63	10.42	1.05 × 10^−6^ *
IL-4	12.52	15.25	22.49	36.25	1.05 × 10^−8^ *
IL-6	1.50	1.98	2.07	3.31	1.10 × 10^−5^ *
IL-10	1.00	1.06	1.05	1.39	0.12
IL-21	10.87	12.33	48.49	33.10	3.57 × 10^−6^ *
APRIL	2073	2072	2868	1924	0.79
BAFF	476.8	545.2	592.8	709.4	2.65 × 10^−4^ *
PBEF	1015	818	1008	1583	0.22
IFN-α	2.65	2.99	5.67	5.68	4.32 × 10^−9^ *
TNF-α	4.44	3.23	4.41	4.46	0.003 *
GM-CSF	1.59	2.32	7.99	5.06	3.35 × 10^−9^ *
NRG-1β1	12.19	20.37	76.04	102.29	1.24 × 10^−4^ *
NGF-β	2.84	1.67	11.33	3.99	3.89 × 10^−8^ *
GDNF	1.56	2.21	2.59	3.39	6.0 × 10^−7^ *
IL-10/IL-6	0.70	0.54	0.51	0.46	0.057
Sex (F/M), %	50/50	78.9/21.1	40/60	61.5/38.5	F vs. M in Low infl. cluster: 0.025 *; F vs. M in High infl. cluster: 0.8
Age	40.00	38.50	34.00	40.00	0.41
Age at disease manifestation	-	25.50	-	25.00	0.75
Disease duration	-	15.50	-	16.50	0.61
PANSS, positive score	-	17.00	-	17.00	0.60
PANSS, negative score	-	22.00	-	18.00	0.25
PANSS, general score	-	42.50	-	33.50	0.10
PANSS, total score	-	82.00	-	66.00	0.17
Total CPZeq	-	300.00	-	279.50	0.18
Antipsychotic type (atypical/typical/polytherapy), %	-	64%/21%/15%	-	57%/9%/34%	-

Note: Data are presented as a median value. The significance of the differences was calculated using the Kruskal–Wallis ANOVA test or Pearson’s chi-squared test (for sex). * Statistically significant differences.

## Data Availability

The data presented in this study are available from the corresponding author on reasonable request.

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
