# Peer review of "Association of Peripheral Inflammatory Biomarkers and Growth Factors Levels with Sex, Therapy and Other Clinical Factors in Schizophrenia and Patient Stratification Based on These Data"

_brainsci, 2023, doi:10.3390/brainsci13050836_

Round 1

Reviewer 1 Report

Ermakov et al. studied multiple inflammatory markers in the serum of patients diagnosed with schizophrenia and compared them to a control group. As a general result, they found increased pro-inflammatory markers in schizophrenia. Further analyses revealed sex differences, specifically a robust proinflammatory state in females. Despite having a modest number of patients, the results of this research combine clinical and biochemical findings on inflammation in schizophrenia, as well as the potential effect of antipsychotics depending on the generation to which they belong.

This interesting and well-designed study provides relevant information on inflammation and schizophrenia. However, there are some issues that authors must address before publishing, which I have listed below:

Major

·       Throughout the manuscript, but not in tables, the authors use gender qualifications (women/men) referring to sex classification (males/females). Correct this issue.

·       As the authors mention, the anti-inflammatory and antioxidant properties of antipsychotics have been described. However, these effects vary from drug to drug. Please provide the full list of antipsychotics that were included in this study.

·       In Table 4 for schizophrenia groups, information about the type of antipsychotic for low and high inflammation clusters is missing. Please provide this information and discuss it.

·       The authors fall short in discussing the information on neurotrophic factors. Exist multiple clinical studies as well as in vitro and animal model reports demonstrating the effect of antipsychotics on neurotrophic factors-related pathways. This may be useful to reinforce this part of the discussion and highlight the potential effect of antipsychotics on these molecules.

·       One interesting finding of this report is that patients exposed to atypical antipsychotics have increased serum levels of pro-inflammatory mediators in comparison with patients under typical antipsychotic treatment. As the authors discuss, this is interesting since it has been widely reported the anti-inflammatory properties of atypical antipsychotics. Subsequently (lines 392-394), the authors state that exists some information about the pro-inflammatory consequences of atypical antipsychotics and highlight clozapine as an example. However, for the information in the methods section, it can be understood that clozapine was not included in the study. Please clarify this (this links with the previous point). Moreover, typical antipsychotics also have anti-inflammatory properties, and this information is missing in this part of the discussion.

Minor

·       In Figure 1 panels J and K please add the alpha Greek letter in the titles.

Author Response

Dear Reviewer,

The authors are grateful for the thoughtful analysis of our manuscript and insightful comments. We have carefully considered and applied your feedback to improve our manuscript. All revisions were highlighted using the "Track Changes" function in Microsoft Word.

Below we answer your suggestions point by point. Please note that your comments are in italics for readability. Our responses are in regular type.

This interesting and well-designed study provides relevant information on inflammation and schizophrenia. However, there are some issues that authors must address before publishing, which I have listed below:

Major

  • Throughout the manuscript, but not in tables, the authors use gender qualifications (women/men) referring to sex classification (males/females). Correct this issue.

Reply: Sorry for this inaccuracy. We have corrected the references to males and females in the text. Thank you for noticing this.

  • As the authors mention, the anti-inflammatory and antioxidant properties of antipsychotics have been described. However, these effects vary from drug to drug. Please provide the full list of antipsychotics that were included in this study.

Reply: We have added information on antipsychotic treatment to the Materials and Methods section.

All patients were taking antipsychotic drugs prescribed by the treating physician including typical antipsychotics: haloperidol, chlorpromazine, trifluoperazin; and atypical antipsychotics: risperidone, quetiapine, olanzapine, clozapine.

  • In Table 4 for schizophrenia groups, information about the type of antipsychotic for low and high inflammation clusters is missing. Please provide this information and discuss it.

Reply: The structure of antipsychotic therapy in clusters of low and high inflammation:

Low inflammation cluster

13 people - typical antipsychotics (54% haloperidol/ 31% trifluoperazin/ 15% chlorprothixene); 22 people - atypical antipsychotics (32% risperidone/ 50% quetiapine or olanzapine/ 18% clozapine); 3 people - mix therapy.

High inflammation cluster

3 people - typical antipsychotics (100% haloperidol); 9 people - atypical antipsychotics (56% risperidone/ 22% quetiapine or olanzapine/ 22% clozapine); 2 people - mix therapy.

Both the low and high inflammation clusters contain approximately 2/3 of patients taking atypical antipsychotics. A proportion of patients taking typical antipsychotics make up 21% and 34% in the low and high inflammation clusters, respectively, which represents comparable proportions, considering the significantly smaller number of patients in the high inflammation cluster. Thus, we believe that the expression of the inflammatory phenotype (cluster) is an inherent characteristic of these patients, despite the differences observed between patients receiving different classes of antipsychotics. However, studies with more sample size of patients receiving monotherapy are needed for more detailed analysis of these effects.

We added additional data to table 4 and comments in the discussion section (please see Table 4 and lines 431-456)

  • The authors fall short in discussing the information on neurotrophic factors. Exist multiple clinical studies as well as in vitro and animal model reports demonstrating the effect of antipsychotics on neurotrophic factors-related pathways. This may be useful to reinforce this part of the discussion and highlight the potential effect of antipsychotics on these molecules.

Reply: Thank you for your valuable feedback. Indeed, the discussion on this issue wasn't presented sufficiently. We have expanded the discussion on this question (please see lines 457-474).

  • One interesting finding of this report is that patients exposed to atypical antipsychotics have increased serum levels of pro-inflammatory mediators in comparison with patients under typical antipsychotic treatment. As the authors discuss, this is interesting since it has been widely reported the anti-inflammatory properties of atypical antipsychotics. Subsequently (lines 392-394), the authors state that exists some information about the pro-inflammatory consequences of atypical antipsychotics and highlight clozapine as an example. However, for the information in the methods section, it can be understood that clozapine was not included in the study. Please clarify this (this links with the previous point). Moreover, typical antipsychotics also have anti-inflammatory properties, and this information is missing in this part of the discussion.

Reply: In our study, among patients taking atypical antipsychotics, 42% received risperidone and 58% received quetiapine, olanzapine, or clozapine.

Indeed, most of works postulated that atypical antipsychotics have some anti-inflammatory properties. Nevertheless, our study showed that in the group of patients taking atypical antipsychotics, the level of pro-inflammatory cytokines was higher than in patients on typical antipsychotics.

We started this discussion of this issue with the pro-inflammatory effects of clozapine (please see lines 431-435). Since clozapine was one of the antipsychotics prescribed to patients, we left this text. Next, we expanded the discussion of this issue and added information about other atypical antipsychotics and typical antipsychotics (please see lines 435-449).

Melbourne et al. [1] showed that risperidone and protracted illness were associated with an increase in JAK-STAT1 signature in peripheral blood mononuclear cells, shifting them to a pro-inflammatory phenotype. We have also previously shown that metabolic syndrome can impact to multidirectional changes of cytokine levels in patients receiving the same therapy with different atypical antipsychotics [2]. In particular, treatment with risperidone, as the least atypical antipsychotic, led to an increase in pro-inflammatory IFN-α2, IL-1β in schizophrenia patients with metabolic syndrome and was ineffective in lowering pro-inflammatory cytokines in patients without it.

 In contrast to the anti-inflammatory effect of many atypical antipsychotics in first administration [3] or short-term drug therapy [4], the extended usage of antipsychotic drugs might lead to metabolic disorders and an increase of inflammatory markers in some cases [5, 6]. Some research from 1990th demonstrated that repeated administration of atypical antipsychotics, i.e., clozapine or risperidone, significantly increase levels of pro-inflammatory cytokines IL-6 and TNFα [7, 8, 9].  Vassilopoulou E. et al. in the long term study of side effects during 5-years of atypical antipsychotics treatment indicated their heterogeneity in metabolic and inflammatory effects [10].

According to in vitro studies, typical antipsychotics, including haloperidol and chlorpromazine, also have anti-inflammatory properties [11, 12, 13, 14]. Obuchowicz E. et al. have demonstrated that chlorpromazine and haloperidol dependently on initial glia activation status via different mechanisms may induce anti-inflammatory effect and decrease production of pro-inflammatory cytokines [14].

Thus, the observed difference in cytokine level changes between typical and atypical antipsiotics is surprising, but may be explained by the described pro-inflammatory effects of atypical antipsychotics and anti-inflammatory effects of typical antipsychotics.

References.

  1. Melbourne JK, Pang Y, Park MR, Sudhalkar N, Rosen C, Sharma RP. Treatment with the antipsychotic risperidone is associated with increased M1-like JAK-STAT1 signature gene expression in PBMCs from participants with psychosis and THP-1 monocytes and macrophages. Int Immunopharmacol. 2020 Feb;79:106093. doi: 10.1016/j.intimp.2019.106093. Epub 2019 Dec 25. PMID: 31863919; PMCID: PMC8792805.
  2. Boiko, A.S.; Mednova, I.A.; Kornetova, E.G.; Gerasimova, V.I.; Kornetov, A.N.; Loonen, A.J.M.; Bokhan, N.A.; Ivanova, S.A. Cytokine Level Changes in Schizophrenia Patients with and without Metabolic Syndrome Treated with Atypical Antipsychotics. Pharmaceuticals 2021, 14, 446.
  3. Noto C, Ota VK, Gouvea ES, Rizzo LB, Spindola LM, Honda PH, Cordeiro Q, Belangero SI, Bressan RA, Gadelha A, Maes M, Brietzke E. Effects of risperidone on cytokine profile in drug-naïve first-episode psychosis. Int J Neuropsychopharmacol. 2014 Oct 31;18(4):pyu042. doi: 10.1093/ijnp/pyu042. PMID: 25522386; PMCID: PMC4360233.
  4. Alvarez-Herrera, S., Escamilla, R., Medina-Contreras, O., Saracco, R., Flores, Y., Hurtado-Alvarado, G., ... & Pavón, L. (2020). Immunoendocrine peripheral effects induced by atypical antipsychotics. Frontiers in endocrinology, 11, 195.
  5. Kelsven, S.; de la Fuente-Sandoval, C.; Achim, C.L.; Reyes-Madrigal, F.; Mirzakhanian, H.; Domingues, I.; Cadenhead, K. Immuno-Inflammatory Changes across Phases of Early Psychosis: The Impact of Antipsychotic Medication and Stage of Illness. Schizophr. Res. 2020, 226, 13–23.
  6. Song X, Fan X, Li X, Zhang W, Gao J, Zhao J, Harrington A, Ziedonis D, Lv L. Changes in pro-inflammatory cytokines and body weight during 6-month risperidone treatment in drug naïve, first-episode schizophrenia. Psychopharmacology (Berl). 2014 Jan;231(2):319-25. doi: 10.1007/s00213-013-3382-4. Epub 2013 Dec 14. PMID: 24337064.
  7. Maes, M., Meltzer, H. Y. (1995). Interleukin-2 and interleukin-6 in schizophrenia and mania: effects of neuroleptics and mood stabilizers. Journal of psychiatric research, 29(2), 141-152. DOI: 10.1016/0022-3956(94)00049-w
  8. Maes, M., Bosmans, E., Kenis, G., De Jong, R., Smith, R. S., & Meltzer, H. Y. (1997). In vivo immunomodulatory effects of clozapine in schizophrenia. Schizophrenia Research, 26(2-3), 221-225. DOI: 10.1016/s0920-9964(97)00057-1
  9. Pollmacher, T., Hinze-Selch, D., & Mullington, J. (1996). Effects of clozapine on plasma cytokine and soluble cytokine receptor levels. Journal of clinical psychopharmacology, 16(5), 403-409. DOI: 10.1097/00004714-199610000-00011
  10. Vassilopoulou E, Efthymiou D, Papatriantafyllou E, Markopoulou M, Sakellariou EM, Popescu AC. Long Term Metabolic and Inflammatory Effects of Second-Generation Antipsychotics: A Study in Mentally Disordered Offenders. J Pers Med. 2021 Nov 12;11(11):1189. doi: 10.3390/jpm11111189. PMID: 34834541; PMCID: PMC8617708.
  11. Patel, S., Keating, B. A., & Dale, R. C. (2022). Anti-inflammatory properties of commonly used psychiatric drugs. Frontiers in Neuroscience, 16. https://doi.org/10.3389/fnins.2022.1039379
  12. Al-Amin, M., Uddin, M. M. N., and Reza, H. M. (2013). Effects of antipsychotics on the inflammatory response system of patients with schizophrenia in peripheral blood mononuclear cell cultures. Clin. Psychopharmacol. Neurosci. 11, 144–151. doi: 10.9758/cpn.2013.11.3.144
  13. Himmerich, H., Bartsch, S., Hamer, H., Mergl, R., Schönherr, J., Petersein, C., et al. (2013). Impact of mood stabilizers and antiepileptic drugs on cytokine production in-vitro. J. Psychiatr. Res. 47, 1751–1759. doi:10.1016/j.jpsychires.2013.07.026
  14. Szuster-Ciesielska, A., Slotwinska, M., Stachura, A., Marmurowska-Michalowska, H., and Kandefer-Szerszen, M. (2004). Neuroleptics modulate cytokine and reactive oxygen species production in blood leukocytes of healthy volunteers. Arch. Immunol. Ther. Exp. 52, 59–67.
  15. Obuchowicz, E., Bielecka-Wajdman, A. M., Paul-Samojedny, M., & Nowacka, M. (2017). Different influence of antipsychotics on the balance between pro-and anti-inflammatory cytokines depends on glia activation: an in vitro study. Cytokine, 94, 37-44. https://doi.org/10.1016/j.cyto.2017.04.004

Minor

  • In Figure 1 panels J and K please add the alpha Greek letter in the titles.

Reply: Figure 1 has been corrected.

Thank you for your thorough analysis of the manuscript and valuable suggestions.

Best regards

Authors

Reviewer 2 Report

Thanks for the interesting and useful article.

Comments:

 1-In the Introduction, talk about the types of psychosis and the difference between schizophrenia and other psychosis(Schizoaffective-Delusional disorder-...),as well as the diagnostic criteria of Schizophrenia based on ICD or DSM.

2- In the methodology, according to the role of endocrine diseases(thyroid, adrenal,...) in psychiatric disorders, have they been managed in the research?

3-How was the use of anti-inflammatory drugs(esp. analgesic agents)managed in patients during research?

4-Considering the difference in the effect of typical and atypical antipsychotic drugs and inflammatory factors, how is this effect managed?

5- It is recommended to generalize the results to all schizophrenic patients with caution, considering the relatively limited number of samples and race.

Author Response

Dear Reviewer,

We thank the reviewer for the positive evaluation of our study, helpful criticisms and valuable suggestions, following which we significantly modified our manuscript. We believe that these changes have improved our paper and clarified our data presentation. All revisions of the manuscript were highlighted using the "Track Changes" function in Microsoft Word.

Below we answer your suggestions point by point. Please note that your comments are in italics and our responses are in regular font for readability.

1-In the Introduction, talk about the types of psychosis and the difference between schizophrenia and other psychosis(Schizoaffective-Delusional disorder-...),as well as the diagnostic criteria of Schizophrenia based on ICD or DSM.

Reply: Since our study was carried out within the framework of the ICD-10, we have added a description of possible symptoms and diagnostic criteria based on this system. In our study, we consider immunological disorders as a potential tool for stratifying patients with schizophrenia into subclasses based on inflammatory phenotype. Therefore, we did not dwell so much on the description of the types of psychoses, but concentrated more on the discussion of the immunological background of the study. However, we have tried to expand the description of the clinical manifestations of schizophrenia in the introduction.

2- In the methodology, according to the role of endocrine diseases(thyroid, adrenal,...) in psychiatric disorders, have they been managed in the research?

Reply: Indeed, some research suggests that individuals with schizophrenia have abnormalities in the hypothalamic-pituitary-adrenal (HPA) axis, which can lead to abnormalities in cortisol levels, and in the thyroid hormone system in individuals with schizophrenia [1, 2, 3]. But, taking into account the long duration of disease of patients with schizophrenia (minimum duration of the disease is 7 years), endocrine disorders as the main cause of negative symptoms or cognitive impairments were excluded in the process of differential diagnosis by psychiatrists. The patients included in the study did not show any clinical signs of endocrine dysfunction of thyroid or adrenal glands. Therefore, the study did not consider the possible effects of any endocrine abnormalities.

In addition, a recent publication in Immunology (2021) by Martin Jaeger et al. described the effects of thyroid and pituitary hormones, thyrotropin and thyroxine, on immune homeostasis in 800 healthy volunteers [4]. The study examined the relationship between TSH and fT4 hormone levels, different circulating cytokines (such as IL-6, IL-1β, IL-1Ra, IL-18, IL-18BP, and VEGF), and immunoglobulin levels. Authors did not detect any associations between hormone levels and ex vivo induced cytokine production, nor any significant associations among hormone levels and cytokine or immunoglobulin levels.

Nevertheless, we have expanded the Limitations section by including information that the potential effect of endocrine and metabolic disturbances was not considered in our study.

References

  1. Goldman MB, Wang L, Wachi C, Daudi S, Csernansky J, Marlow-O'Connor M, Keedy S, Torres I. Structural pathology underlying neuroendocrine dysfunction in schizophrenia. Behav Brain Res. 2011 Mar 17;218(1):106-13. doi: 10.1016/j.bbr.2010.11.025. Epub 2010 Nov 17. PMID: 21093493; PMCID: PMC4465073.
  2. Yang, K., Zeng, X., Hu, L., Chen, J., Zhu, W., Wang, Y., & Zhao, X. (2023). Magnetic Resonance Imaging Characteristics of Brain Structure and Neuroendocrine Changes in Patients with First-Episode Schizophrenia. Concepts in Magnetic Resonance Part A, Bridging Education and Research, 2023.
  3. Lally, J., Sahl, A. B., Murphy, K. C., Gaughran, F., & Stubbs, B. (2019). Serum prolactin and bone mineral density in schizophrenia: a systematic review. Clinical Psychopharmacology and Neuroscience, 17(3), 333.
  4. Jaeger M, Sloot YJE, Horst RT, Chu X, Koenen HJPM, Koeken VACM, Moorlag SJCFM, de Bree CJ, Mourits VP, Lemmers H, Dijkstra H, Medici M, van Herwaarden AE, Joosten I, Joosten LAB, Li Y, Smit JWA, Netea MG, Netea-Maier RT. Thyrotrophin and thyroxine support immune homeostasis in humans. Immunology. 2021 Jun;163(2):155-168. doi: 10.1111/imm.13306. Epub 2021 Feb 7. PMID: 33454989; PMCID: PMC8114202. https://onlinelibrary.wiley.com/doi/full/10.1111/imm.133063-How was the use of anti-inflammatory drugs(esp. analgesic agents)managed in patients during research?

Reply: We did not consider the influence of analgesics in the study as such data were not available to us. We have added this note to the limitations of the study.

4-Considering the difference in the effect of typical and atypical antipsychotic drugs and inflammatory factors, how is this effect managed?

Reply: In our study, surprisingly, the results indicate an increase in certain pro-inflammatory cytokines in patients taking atypical antipsychotics. We have expanded the discussion of these results and the effects of different classes of antipsychotics on cytokine and neurotrophic factor levels in the discussion section and possible limitations in the interpretation of these data in the limitations section.

5- It is recommended to generalize the results to all schizophrenic patients with caution, considering the relatively limited number of samples and race.

Reply: We agree that the results obtained should be interpreted with caution. Therefore, we added subsection 4.4. Significance of Findings, Limitations and Directions for Future Research to the Discussion section. We have added information about the limited sample size. We have made changes to the text of the discussion and we have reformulated the conclusion.

Thank you for your thorough analysis of our manuscript.

Best regards

Authors

Reviewer 3 Report

Overview

Thank you for the opportunity to review this work. This work was a cross-sectional comparison of serum inflammatory biomarkers (including cytokines and growth factors) between schizophrenia patients and healthy controls. The main aims seem to be comparing biomarker levels between groups and examining relationships with clinical symptoms and characteristics. However, the manuscript still has the following problems worthy of attention, through the improvement of these problems can better improve the quality of the manuscript.

Abstract

The Abstract should be concise while covering the key elements, which include the research objectives/hypotheses, methods, main findings, and conclusions. The current Abstract contains too much detail and would benefit from pruning. Remove details such as "fifteen biomarkers including" and just state the number. Avoid excessive specifications on statistical significance "p<0.05".

Use parallel grammatical forms for conciseness, e.g. "IL-6 was higher; IL-10 was lower" rather than "There was an increase in IL-6; IL-10 levels showed a decrease". Remove excess wording like "lines of evidence are known" and "active involvement of".

Remove excessive details on subgroup analyses and clinical correlates. Focus on the key significant results relating to group differences and cluster analysis. Additional details can be moved to the main manuscript.

Proofread carefully. The Abstract should be coherent and flow logically from one idea to the next. Remove any repetition or wordiness. Ensure consistent terms and parallel sentence structure.

Introduction

Clearly state the research objectives and hypotheses at the end of the introduction to guide the reader. For example, "Therefore, the current study aimed to 1) analyze the changes in a panel of cytokines and growth factors in patients with schizophrenia compared to healthy controls; 2) explore the effects of clinical variables on biomarker levels; 3) subgroup patients based on inflammatory markers and compare their clinical profiles."

Use more topic sentences to enhance the coherence and logic. For example, add a topic sentence for the first paragraph "Schizophrenia is associated with immune dysregulation and inflammation."

Avoid abrupt ending. Add a concluding sentence to wrap up the introduction smoothly. For example, "This study investigated serum inflammatory and neurotrophic markers in schizophrenia to gain further insights into the role of immune and growth factors in schizophrenia pathogenesis and heterogeneity."

Pay attention to the overall structure and logical flow. The current structure seems a bit abrupt and scattered.

Methods

Authors could subdivide the section into clear subsections for better organization and flow, including Study Design, Participants, Procedure, Materials, and Statistical Analysis.

In the Participants subsection, report the inclusion and exclusion criteria in a list format for better clarity.

In the Procedure subsection, describe the steps for data collection and laboratory testing in a chronological order. For example, “Fasting blood samples were collected...Serum was extracted by centrifugation and stored...Biomarker levels were measured using Luminex multiplex assay”. Provide more specifics on the procedures.

In the Materials subsection, report details on the source and catalog number of reagents and instruments.

In the Statistical Analysis subsection, describe the statistical tests in a separate paragraph for each analysis.

Provide more details on the cluster analysis, including the clustering algorithm, distance metric, number of clusters, and cluster assignment.

Include a data preprocessing subsection to report how missing data and outliers were handled.

Results

Add a preamble paragraph briefly restating the study aims and hypotheses/objectives to orient the readers before presenting the results. For example, "To investigate immune dysregulation and inflammation in schizophrenia, we compared serum levels of cytokines and growth factors between schizophrenia patients and healthy controls. We also examined associations between biomarker levels and demographic/clinical characteristics. Finally, we performed cluster analysis to identify subgroups with high vs. low inflammation."

Use headings and subheadings to improve the logical flow and clarity. For example, 3.1 Demographic and Clinical Characteristics, 3.2 Group Differences in Biomarker Levels, 3.3 Correlations with Demographic/Clinical Variables, 3.4 Cluster Analysis.

Correlation analyses can also be presented in a table indicating the correlation coefficient and p value for each biomarker and demographic/clinical variable. Visualize any significant correlations using scatterplots.

For the cluster analysis, provide more details on the clustering method, number of clusters, and cluster sizes.

Discussion

The Discussion lacks a clear flow or structure. Use headings and subheadings to organize your discussion around themes or topics, e.g. group differences in biomarkers, demographic/clinical correlates, cluster analysis, implications/future directions. 

Start the Discussion with a summary of the key results and main findings from the study before comparing with previous literature. Briefly restate the aims and hypotheses. For example, "The present study aimed to compare serum inflammatory biomarkers between schizophrenia patients and controls, and examine associations with demographic/clinical characteristics. We found significantly higher levels of pro-inflammatory cytokines like IL-6 and lower IL-10/IL-6 ratios in schizophrenia patients, indicating an inflammatory state. Around 25% of patients showed a 'high inflammation' profile based on cluster analysis."

When comparing results with previous studies, use topic sentences to indicate which finding you are comparing. Discuss both consistent and inconsistent results, and potential reasons for inconsistencies. Reference relevant studies to support each conclusion. For example, "Our finding of increased IL-6 in schizophrenia is consistent with multiple previous studies (refs). In contrast, we found decreased TNF-α levels, unlike most other studies reporting no change or increased TNF-α (refs). The decrease in our sample may be explained by the higher proportion of males, where TNF-α was significantly lower."

Expand on the demographic and clinical correlates, including both significant and non-significant findings. Discuss potential explanations and implications. For example, "We did not find significant associations between biomarker levels and age, illness duration or PANSS scores, possibly due to limited sample size. However, patients with predominantly negative symptoms showed higher IL-6 and lower IL-10/IL-6 ratios, indicating a link between inflammation and negative symptom severity that warrants further investigation."

Discuss the cluster analysis in more depth. How do the 'low inflammation' and 'high inflammation' subgroups differ in terms of biomarkers and demographic/clinical characteristics? What are the implications for developing inflammatory-based subgroups or personalized treatment approaches? For example, "The 'high inflammation' cluster comprised around 25% of patients, with significantly higher levels of both pro-inflammatory cytokines as well as certain growth factors like NRG-1β1. Though clinical parameters did not differ significantly between clusters, the 'high inflammation' group may represent an inflammatory endophenotype of schizophrenia with distinct pathophysiology and treatment needs."

End the Discussion with a summary of the strengths/limitations, significance/implications of findings, and directions for future research. For example, "Our study provides further evidence for immune system dysregulation and inflammatory abnormalities in a subgroup of schizophrenia patients. The identification of inflammatory biomarkers may aid in diagnosis, prognostication and guiding personalized treatment approaches. Larger longitudinal studies are needed to confirm these findings, and determine how inflammatory profiles may relate to disease manifestation, severity, treatment response and cognition."

Conclusions

The current conclusions are very broad and lack specificity. Rewrite the conclusions to focus on the key significant findings and main takeaway messages from your study. Avoid generic statements. For example, instead of "Patients were characterized by a clear pro-inflammatory phenotype.", you can state "We found significantly higher levels of the pro-inflammatory cytokine IL-6 and lower anti-inflammatory IL-10 levels in schizophrenia patients, indicating a pro-inflammatory state."

The language comes across as somewhat wordy, repetitive and speculative at times. There are lengthy, complex sentences that could be more concise while still retaining meaning. I would recommend simplifying sentence structure, removing unnecessary words/phrases and qualifying speculations when possible. Focus on being clear and concise.

The tone seems inconsistent, ranging from formal to colloquial. For an academic research paper, a formal and objective tone is most appropriate. I would avoid colloquial expressions, first person pronouns (we/our) and focus on using impartial language to present the research.

Transitional phrases are used at times but could be employed more regularly to create better flow and coherence, especially between paragraphs. Additional transitional words/phrases for comparison, causation, illustration, etc. would help guide the reader.

Headings and subheadings are present but could be improved. Headings should clearly indicate the main focus or theme of each section. Subheadings can then guide the reader through more specific ideas. I would suggest revising some headings to be more parallel and concise.

Terminology and style seem generally consistent but there are minor inconsistencies in things like abbreviations, technical terms, and phrasing. Carefully proofreading to ensure consistency with style, formatting, terminology, etc. throughout the manuscript would strengthen quality.

Overall, the language style is moderately academic but would benefit from additional polishing to be concise yet compelling, with objective and consistent writing.

Author Response

Dear Reviewer,

The authors are deeply grateful for the thoughtful analysis of our manuscript and valuable suggestions. This is the first time we have received such useful feedback on our article. We believe that your suggestions have improved the manuscript significantly. Moreover, we find useful your suggestions for our other research articles. Many of your suggestions will be valuable to young scientists who want to improve the quality of results presentation.

Below we answer your suggestions point by point. All revisions of the manuscript were highlighted using the "Track Changes" function in Microsoft Word. Please note that your comments are in italics and our responses are in regular font for readability.

Overview

Thank you for the opportunity to review this work. This work was a cross-sectional comparison of serum inflammatory biomarkers (including cytokines and growth factors) between schizophrenia patients and healthy controls. The main aims seem to be comparing biomarker levels between groups and examining relationships with clinical symptoms and characteristics. However, the manuscript still has the following problems worthy of attention, through the improvement of these problems can better improve the quality of the manuscript.

Abstract

The Abstract should be concise while covering the key elements, which include the research objectives/hypotheses, methods, main findings, and conclusions. The current Abstract contains too much detail and would benefit from pruning. Remove details such as "fifteen biomarkers including" and just state the number. Avoid excessive specifications on statistical significance "p<0.05".

Reply: Thanks for the suggestion. We removed the phrases you mentioned from the abstract.

Use parallel grammatical forms for conciseness, e.g. "IL-6 was higher; IL-10 was lower" rather than "There was an increase in IL-6; IL-10 levels showed a decrease". Remove excess wording like "lines of evidence are known" and "active involvement of".

Reply: We rewrote the sentence you mentioned and removed some redundant words from the abstract.

Remove excessive details on subgroup analyses and clinical correlates. Focus on the key significant results relating to group differences and cluster analysis. Additional details can be moved to the main manuscript.

Reply: We believe that the results of subgroup analyzes and clinical correlates are an important part of our work, so we describe this in the abstract. However, we have removed the details and reduced the description of these results to 2 sentences.

Proofread carefully. The Abstract should be coherent and flow logically from one idea to the next. Remove any repetition or wordiness. Ensure consistent terms and parallel sentence structure.

Reply: We have reduced the Abstract significantly and proofread carefully. We also fixed the parallel structure of some sentences. In our opinion, the Abstract has become more coherent and logical.

Introduction

Clearly state the research objectives and hypotheses at the end of the introduction to guide the reader. For example, "Therefore, the current study aimed to 1) analyze the changes in a panel of cytokines and growth factors in patients with schizophrenia compared to healthy controls; 2) explore the effects of clinical variables on biomarker levels; 3) subgroup patients based on inflammatory markers and compare their clinical profiles."

Reply: We have changed the last paragraph and added your proposed statement of the aim of this work.

Use more topic sentences to enhance the coherence and logic. For example, add a topic sentence for the first paragraph "Schizophrenia is associated with immune dysregulation and inflammation."

Reply: A sentence with a similar meaning was already in the first paragraph. But we've added a suggested phrase.

Avoid abrupt ending. Add a concluding sentence to wrap up the introduction smoothly. For example, "This study investigated serum inflammatory and neurotrophic markers in schizophrenia to gain further insights into the role of immune and growth factors in schizophrenia pathogenesis and heterogeneity."

Reply: Thank you for the suggested phrase. We've added it to the Introduction. Concluding sentences has been added to some of the paragraphs.

Pay attention to the overall structure and logical flow. The current structure seems a bit abrupt and scattered.

Reply:  We have revised the Introduction structure. Some introductory words and concluding sentences have been added.

Methods

Authors could subdivide the section into clear subsections for better organization and flow, including Study Design, Participants, Procedure, Materials, and Statistical Analysis.

Reply:  We have reformatted some subsections in the Methods section. But we think the Procedure subsection is too general, so we didn't include it in the text. Instead, we added 2 sections: 2.3. Biological Material and 2.4. Multiplex Immunoassay of Cytokines and Growth Factors in Serum. In addition, the Materials section is also redundant, because all the materials used in the work are listed in the text. The remaining sections were already present in the manuscript.

In the Participants subsection, report the inclusion and exclusion criteria in a list format for better clarity.

Reply:  Inclusion and exclusion criteria were presented in a list format.

In the Procedure subsection, describe the steps for data collection and laboratory testing in a chronological order. For example, “Fasting blood samples were collected...Serum was extracted by centrifugation and stored...Biomarker levels were measured using Luminex multiplex assay”. Provide more specifics on the procedures.

Reply:  We think the Procedure subsection is too general. Typically, research articles indicate specific methods of analysis, rather than a general section with procedures. Therefore, we added one and left one subsection: 2.3. Biological Material and 2.4. Multiplex Immunoassay of Cytokines and Growth Factors in Serum.

In the Materials subsection, report details on the source and catalog number of reagents and instruments.

Reply:  We have not included the Materials subsection, so catalog numbers and manufacturer for reagents are listed in the text in the relevant sections.

In the Statistical Analysis subsection, describe the statistical tests in a separate paragraph for each analysis.

Reply:  The manuscript already includes a paragraph describing the statistical tests used for the analysis. For example, see the following sentence: “The significance of the differences in biomarker levels was calculated using the Mann–Whitney U test or Kruskal–Wallis test followed with Dunn's post hoc test for multiple comparisons (when comparing more than two groups)”.

Provide more details on the cluster analysis, including the clustering algorithm, distance metric, number of clusters, and cluster assignment.

Reply:  Information about the cluster algorithm has already been presented in a separate paragraph. But we have added distance metric information. Euclidean distance is used as distance metric. The optimal number of clusters was estimated using hierarchical clustering. The optimal number of clusters was two. Therefore, further all participants were divided into two clusters using K-means clustering.

Include a data preprocessing subsection to report how missing data and outliers were handled.

Reply:  Information about data preprocessing is presented in the subsection 2.5. Data Preprocessing and Statistical Analysis (please see first paragraph).

Results

Add a preamble paragraph briefly restating the study aims and hypotheses/objectives to orient the readers before presenting the results. For example, "To investigate immune dysregulation and inflammation in schizophrenia, we compared serum levels of cytokines and growth factors between schizophrenia patients and healthy controls. We also examined associations between biomarker levels and demographic/clinical characteristics. Finally, we performed cluster analysis to identify subgroups with high vs. low inflammation."

Reply: In our opinion, such a preamble is redundant. The purpose of the work has already been described twice in the text of the manuscript (at the end of the Introduction section and in subsection 2.1. Study design). There is little point in repeating the same thing a third time. Such a preamble makes sense in the Discussion section.

Use headings and subheadings to improve the logical flow and clarity. For example, 3.1 Demographic and Clinical Characteristics, 3.2 Group Differences in Biomarker Levels, 3.3 Correlations with Demographic/Clinical Variables, 3.4 Cluster Analysis.

Reply: The Results section already contains subsections. At your suggestion, we have changed the wording of some subsections. Thank you for this suggestion.

Correlation analyses can also be presented in a table indicating the correlation coefficient and p value for each biomarker and demographic/clinical variable. Visualize any significant correlations using scatterplots.

Reply: We did a correlation analysis but found no clinically significant correlations except for GDNF level and age (Rs = -0.22, p = 0.04, Spearman correlation). Scatterplots have been presented in Supplementary Figure S1. Presenting a complete table with correlations and p values seems to us redundant. But we have added the following description of correlation analysis to the manuscript: “Correlation analysis did not reveal clinically significant correlations with the analyzed biomarkers, with the exception of age. Serum GDNF concentration decreased with increasing age (Rs = -0.22, p = 0.04, Spearman correlation). This dependence is graphically represented in Supplementary Figure S1”.

For the cluster analysis, provide more details on the clustering method, number of clusters, and cluster sizes.

Reply:  We have added more information about the clustering method. In particular, the following two sentences have been added: “Preliminary cluster analysis using hierarchical clustering showed that the participants are divided into two main clusters. Therefore, further all participants were divided into two clusters using K-means clustering”. However, the software (STATISTICA 10) used for cluster analysis does not provide data on cluster sizes, therefore, we were unable to present these data.

Discussion

The Discussion lacks a clear flow or structure. Use headings and subheadings to organize your discussion around themes or topics, e.g. group differences in biomarkers, demographic/clinical correlates, cluster analysis, implications/future directions.

Reply:  Thank you for this suggestion. Indeed, the addition of subheadings improves the structure and readability of the Discussion section. We have divided the Discussion section into 4 subsections.

Start the Discussion with a summary of the key results and main findings from the study before comparing with previous literature. Briefly restate the aims and hypotheses. For example, "The present study aimed to compare serum inflammatory biomarkers between schizophrenia patients and controls, and examine associations with demographic/clinical characteristics. We found significantly higher levels of pro-inflammatory cytokines like IL-6 and lower IL-10/IL-6 ratios in schizophrenia patients, indicating an inflammatory state. Around 25% of patients showed a 'high inflammation' profile based on cluster analysis."

Reply: Thank you for this suggestion. We've added your suggested short summary of the key results to the top of the discussion.

When comparing results with previous studies, use topic sentences to indicate which finding you are comparing. Discuss both consistent and inconsistent results, and potential reasons for inconsistencies. Reference relevant studies to support each conclusion. For example, "Our finding of increased IL-6 in schizophrenia is consistent with multiple previous studies (refs). In contrast, we found decreased TNF-α levels, unlike most other studies reporting no change or increased TNF-α (refs). The decrease in our sample may be explained by the higher proportion of males, where TNF-α was significantly lower."

Reply: We have tried to add topical sentences to indicate the results being compared. Suggested phrases have been added to the text.

Expand on the demographic and clinical correlates, including both significant and non-significant findings. Discuss potential explanations and implications. For example, "We did not find significant associations between biomarker levels and age, illness duration or PANSS scores, possibly due to limited sample size. However, patients with predominantly negative symptoms showed higher IL-6 and lower IL-10/IL-6 ratios, indicating a link between inflammation and negative symptom severity that warrants further investigation."

Reply: Thank you for this suggestion. We've added the suggested phrase to a subsection 4.2.

Discuss the cluster analysis in more depth. How do the 'low inflammation' and 'high inflammation' subgroups differ in terms of biomarkers and demographic/clinical characteristics? What are the implications for developing inflammatory-based subgroups or personalized treatment approaches? For example, "The 'high inflammation' cluster comprised around 25% of patients, with significantly higher levels of both pro-inflammatory cytokines as well as certain growth factors like NRG-1β1. Though clinical parameters did not differ significantly between clusters, the 'high inflammation' group may represent an inflammatory endophenotype of schizophrenia with distinct pathophysiology and treatment needs."

Reply: We have added subsection “4.3. Patient Stratification by Inflammatory Biomarkers” and slightly expanded the description of cluster analysis results.

End the Discussion with a summary of the strengths/limitations, significance/implications of findings, and directions for future research. For example, "Our study provides further evidence for immune system dysregulation and inflammatory abnormalities in a subgroup of schizophrenia patients. The identification of inflammatory biomarkers may aid in diagnosis, prognostication and guiding personalized treatment approaches. Larger longitudinal studies are needed to confirm these findings, and determine how inflammatory profiles may relate to disease manifestation, severity, treatment response and cognition."

Reply: We have added a subsection “4.4. Significance of Findings, Limitations and Directions for Future Research”.

Conclusions

The current conclusions are very broad and lack specificity. Rewrite the conclusions to focus on the key significant findings and main takeaway messages from your study. Avoid generic statements. For example, instead of "Patients were characterized by a clear pro-inflammatory phenotype.", you can state "We found significantly higher levels of the pro-inflammatory cytokine IL-6 and lower anti-inflammatory IL-10 levels in schizophrenia patients, indicating a pro-inflammatory state."

Reply: We rewrote the conclusion and added the suggested phrases.

Comments on the Quality of English Language

The language comes across as somewhat wordy, repetitive and speculative at times. There are lengthy, complex sentences that could be more concise while still retaining meaning. I would recommend simplifying sentence structure, removing unnecessary words/phrases and qualifying speculations when possible. Focus on being clear and concise.

Reply: We have tried to change the style of the sentences where possible.

The tone seems inconsistent, ranging from formal to colloquial. For an academic research paper, a formal and objective tone is most appropriate. I would avoid colloquial expressions, first person pronouns (we/our) and focus on using impartial language to present the research.

Reply: We tried to change the tone of the sentences where possible.

Transitional phrases are used at times but could be employed more regularly to create better flow and coherence, especially between paragraphs. Additional transitional words/phrases for comparison, causation, illustration, etc. would help guide the reader.

Reply: Transitional phrases have been added to create better flow and coherence.

Headings and subheadings are present but could be improved. Headings should clearly indicate the main focus or theme of each section. Subheadings can then guide the reader through more specific ideas. I would suggest revising some headings to be more parallel and concise.

Reply: We have significantly revised the headings and subheadings especially in the Discussion section.

Terminology and style seem generally consistent but there are minor inconsistencies in things like abbreviations, technical terms, and phrasing. Carefully proofreading to ensure consistency with style, formatting, terminology, etc. throughout the manuscript would strengthen quality.

Reply: We checked terminology, abbreviations, technical terms, and phrasing.

Overall, the language style is moderately academic but would benefit from additional polishing to be concise yet compelling, with objective and consistent writing.

Reply: We have tried to fix the language style where possible.

You have done a great job on our manuscript. Thank you for your help in improving the quality of the presentation.

We were pleased to receive such feedback.

Thanks again for the really valuable suggestions.

Best regards

Authors

Round 2

Reviewer 1 Report

The authors have made the recommendations.

Author Response

Thank you for the fruitful discussion and insightful analysis of our manuscript.